# Introductory Engineering Mathematics Students' Weighted Score Predictions Utilising a Novel Multivariate Adaptive Regression Spline Model

**Abul Abrar Masrur Ahmed** [1,2], **Ravinesh C. Deo** [1,*], **Sujan Ghimire** [1], **Nathan J. Downs** [1], **Aruna Devi** [3], **Prabal D. Barua** [4] and **Zaher M. Yaseen** [1,5,6]

1   UniSQ's Advanced Data Analytics Research Group, School of Mathematics, Physics, and Computing, University of Southern Queensland, Springfield, QLD 4300, Australia
2   Department of Infrastructure Engineering, The University of Melbourne, Parkville, VIC 3010, Australia
3   School of Education and Tertiary Access, The University of the Sunshine Coast, Caboolture, QLD 4510, Australia
4   School of Business, University of Southern Queensland, Springfield, QLD 4300, Australia
5   New Era and Development in Civil Engineering Research Group, Scientific Research Center, Al-Ayen University, Thi-Qar 64001, Iraq
6   Institute for Big Data Analytics and Artificial Intelligence (IBDAAI), Kompleks Al-Khawarizmi, Universiti Teknologi MARA, Shah Alam 40450, Selangor, Malaysia
*   Correspondence: ravinesh.deo@usq.edu.au; Tel.: +61-07-3470-4430

**Abstract:** Introductory Engineering Mathematics (a skill builder for engineers) involves developing problem-solving attributes throughout the teaching period. Therefore, the prediction of students' final course grades with continuous assessment marks is a useful toolkit for degree program educators. Predictive models are practical tools used to evaluate the effectiveness of teaching as well as assessing the students' progression and implementing interventions for the best learning outcomes. This study develops a novel multivariate adaptive regression spline (MARS) model to predict the weighted score *WS* (i.e., the course grade). To construct the proposed MARS model, Introductory Engineering Mathematics performance data over five years from the University of Southern Queensland, Australia, were used to design predictive models using input predictors of online quizzes, written assignments, and examination scores. About 60% of randomised predictor grade data were applied to train the model (with 25% of the training set used for validation) and 40% to test the model. Based on the cross-correlation of inputs vs. the *WS*, 12 distinct combinations with single (i.e., M1–M5) and multiple (M6–M12) features were created to assess the influence of each on the *WS* with results bench-marked via a decision tree regression (DTR), kernel ridge regression (KRR), and a *k*-nearest neighbour (KNN) model. The influence of each predictor on *WS* clearly showed that online quizzes provide the least contribution. However, the MARS model improved dramatically by including written assignments and examination scores. The research demonstrates the merits of the proposed MARS model in uncovering relationships among continuous learning variables, which also provides a distinct advantage to educators in developing early intervention and moderating their teaching by predicting the performance of students ahead of final outcome for a course. The findings and future application have significant practical implications in teaching and learning interventions or planning aimed to improve graduate outcomes in undergraduate engineering program cohorts.

**Keywords:** educational decision making; multivariate regression spline model; student performance; artificial intelligence in education; engineering mathematics student performance

## 1. Introduction

Predictive modelling can help engineering educators to design an optimal learning and teaching practices considering the feedback generated through student performance data. In terms of monitoring student learning through problem solving, such models

can convey important information on continuous progress and advancement in students' knowledge [1]. Predictive methods [2,3] are therefore a key component of learning analytics methods for dynamic learning environments to encourage students to participate continuously in learning and teaching platforms (both in-classrooms and online) while enabling the educators to evaluate their practice [4]. Based on such models, teaching and learning have progressed dramatically, making them flexible and adaptable [5]. This can help educators to present useful feedback and associated comments on continuous assessments with a description of areas where students are excelling and specific areas that require improvement. Most datasets comprising marks in short tests, online quizzes, or assignments are mathematically and statistically expressible and therefore can be utilised as inputs into learning analytics models with adaptive methods to forecast student progress in a course and thus continuously improve teaching and learning practice.

The literature has not fully established the adoption of statistical and machine learning predictive models, e.g., [2,3], to inform student's short-term future progress through a teaching semester. The present study aims to develop a student progress-monitoring model that can be used as a vital part of the e-teaching and e-learning systems that guide educators in making better decisions to improve their practice for optimal outcomes. Based on the predicted performance using any current assessment, the educators can further implement comprehensive changes in their subsequent assignment or other tasks to capture the potential impact on the weighted course performances and final grades. If predictions are possible, the model can help facilitate a greater understanding of how the performance in continuous assessments improves a final grade and further identify factors that influence the knowledge domain and course progress using different kinds of learning attributes (e.g., online quizzes and written assignments).

In general, continuous assessment variables are based on formative assessments that can determine a student's level of achievement in terms of their evolving capabilities [6,7]. They may also comprise both summative and formative evaluations that form the two kinds of student assessment procedures [8] for generating information regarding student progression before, during, or after any particular set of sequenced learning activities [9]. This information can enable educators to improve learning outcomes [10] and predict student performance as an essential component of a robust education system [11,12]. Considering these benefits, teachers can evaluate and improve their teaching and students' learning processes [13] with subject-specific and general qualities when modelling students' overall performances in a stage-by-stage approach. Subject-specific attributes, for example, can be used to determine how far students may develop in their mastery of various learning materials.

In terms of the published literature, the maximum likelihood estimation method has been used to measure student knowledge levels regarding the difficulty in understanding course learning materials. In another study, students' self-assessment skills were investigated by determining the reasons for a student's failure to solve a problem [14]. This system gathered data on student development primarily based on the difficulty levels and the problem categories. The study of [15] used self-assessment tests to improve students' examination performance, where exam questions were adaptively created based on students' responses to each previously answered question. It was therefore shown that the student's likelihood to answer the questions correctly could be predicted based on their knowledge levels using the item response theory and that the accuracy of the responses and their probability distributions, i.e., the probability of the appropriate knowledge level, in terms of concepts, were also used to grade the students.

Current studies have used classification, and regression approaches such as, but not limited to, support vector machines, decision trees, artificial neural networks, and adaptive neuro-fuzzy inference systems to predict student course performance [16–20]. For example, [21] determined optimal variables to represent student attributes by developing an efficient model to aid in clustering students into distinct groups considering performance levels, behaviour, and engagement. The study of [22] proposed a SPRAR (students' per-

formance prediction using relational association rules) classification model to predict the final result of a student at a certain academic discipline using relational association rules (RARs), conducting experiments performed on three real academic datasets to show its superiority. A study by Goga et al. [23] designed an intelligent recommender system using background factors to predict students' first-year academic performance while recommending actions for improvement, whereas Fariba [24] studied the academic performance of online students using personality traits, learning styles, and psychological wellbeing data, showing a correlation between personality traits and learning styles. It was noted that this could lead learners to a higher level of learning and a sense of self-satisfaction and enjoyment of the learning process. Taylan and Karagözoglu [25] introduced a fuzzy inference system model to assess students' academic performance, showing that their method could produce crisp numerical outcomes to predict student's academic performance and an alternative solution to address imprecise data issues. The study of Ashraf et al. [26] developed base classifiers such as random tree, kernel ridge regression, and Naïve Bayes methods evaluated on a 10-fold validation with filtering such as oversampling (SMOTE) and undersampling (spread subsampling) to inspect any significant change in results among meta and base classifiers. Their study showed that both ensemble and filtering approaches met substantial improvement margins in predicting students' performance compared with conventional classifiers.

Applying classification and prediction methods, Pallathadka et al. [27] developed Naive Bayesian ID3, C4.5, and SVM models on student performance data to forecast student performance, classify individuals based on talents, and enhance future test performance. Other studies, e.g., [28,29], used predicted students' performance in massive open online courses (MOOCs) to study students' retention and make timely interventions and an early prediction of an university undergraduate student's academic performance in completely online learning. The former proposed a hyper-model using convolutional neural network and a long short-term memory model to automatically extract features from MOOCs raw data and to determine course dropout rates, whereas the latter considered a cost-sensitive loss function to study various mis-classification costs for false negatives and false positives.

The study of Deo et al. [3] has developed extreme learning machine models to analyse patterns embedded in continuous assessment to model the weighted course result and examination score for both mid-level (engineering mathematics) and advanced engineering mathematics performance in on-campus and online study modes compared with random forest and Volterra models. Using a statistical approach, Nguyen-Huy et al. [2] developed a probabilistic model to predict weighted scores for on-campus and online students in advanced engineering mathematics. This study fitted parametric and non-parametric D-vine copula models utilising online quizzes, assignments, and examination results to model the predicted course weighted score. This was interpreted as the probability of whether a student's continuous performance, individually or jointly with other assessments, leads to passing course grade conditional upon joint performance in online quizzes and written assignments. Other researchers, such as [20,30,30–41], have attempted to develop several types of classification and regression models and statistical methods for student performance predictions using a diverse set of predictor variables. Despite their success, no single machine learning or statistical model appears to generate universally accurate performance for the diverse datasets representing student performance; therefore, individual differences among these predictive models and the associated contextual factors could be considered when predicting student course performance.

This research builds upon earlier research involving undergraduate university mathematics courses [2,3]. The primary contributions are to develop a novel multivariate adaptive regression spline (MARS) model that has feature identification and regression capabilities to explore relationships between assessment-based predictors and the target course grade outcome. The performance of the proposed MARS model is also benchmarked with $k$-nearest neighbour algorithm (KNN), kernel ridge regression (KRR), and decision tree regression (DTR) using five consecutive years of undergraduate student performance

datasets (2015–2019) for both online and on-campus modes of course offers. The novelty of this research work is to develop a MARS for the first time to predict the first-year engineering mathematics student performance at the University of Southern Queensland, Australia, by employing several continuous assessment marks and the weighted scores that are used to assign a passing or a failing grade. The remainder of the research is dedicated to describing the novel properties of MARS with respect to related benchmark models. Several challenges after the presentation of results are then discussed, and a final section summarizes the conclusions.

## 2. Theoretical Overview and Methodology

### 2.1. Objective Model: Multivariate Adaptive Regression Splines (MARS)

This research presents a MARS model considering multivariate data (online quizzes, assignments, and examination scores) for a first-year undergraduate engineering mathematics course as predictors to emulate weighted scores by analysing the contribution from basis functions derived from each feature. Figure 1 shows the schematic structure of the proposed MARS model.

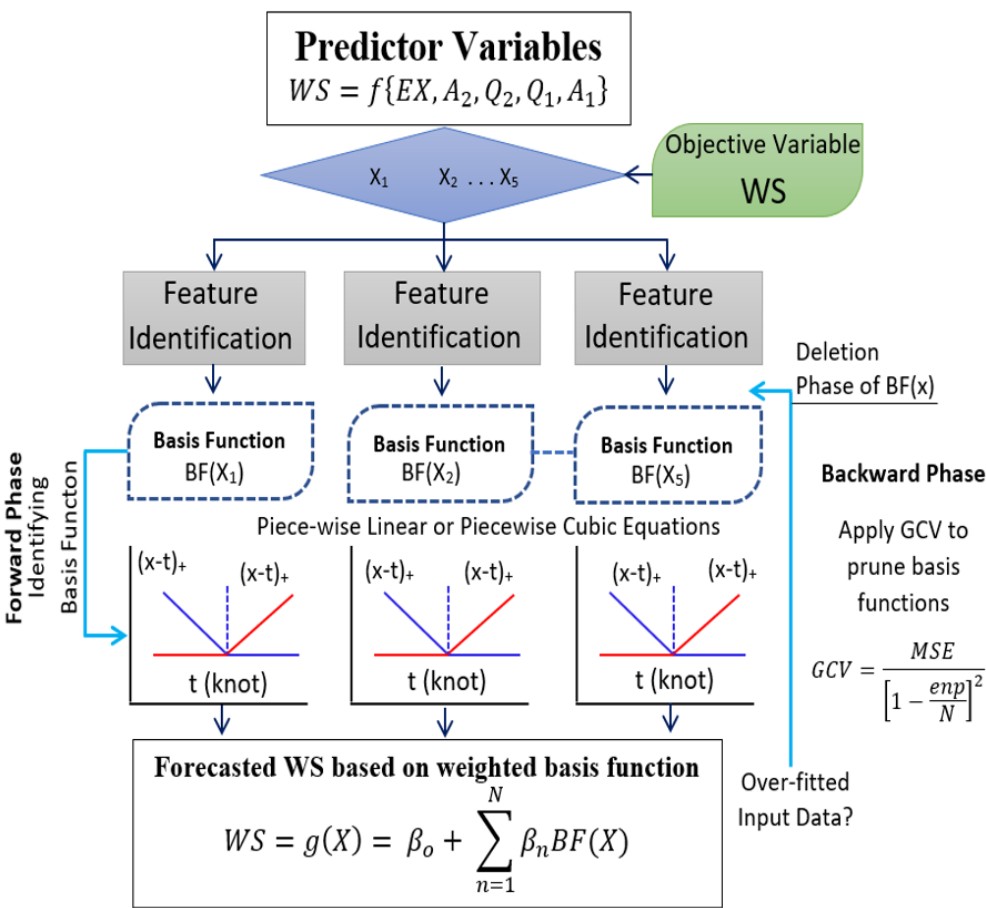

**Figure 1.** The architecture of the newly proposed multivariate adaptive regression splines (MARS) model used to predict undergraduate Introductory Engineering Mathematics student performance at the University of Southern Queensland, Australia.

In this study, a MARS model is selected based on its excellent capability to evaluate the interactive effects of various inputs used to estimate the given predictand variable [42]. Furthermore, this particular model to determine the relative importance of any single predictor (e.g., assignment score) or a combination of predictors (e.g., assignment + quiz marks) on the weighted score, therefore enabling the educator to explore the complex and somewhat non-linear relationships [2,3] based on which an assessment can affect

(and be used to model) a final grade. In the present problem of predicting student course performance (i.e., weighted scores), the proposed MARS model aims to select the most optimal regressor variable such as the online quiz or the written assignment mark from a predictor matrix set [43] without any assumptions on relationships between the respective predictors and the predictand [44,45]. Using such input features, the proposed MARS model generates a forecasted weighted score through the learned relationships represented in the spline function. Each spline, while utilising an input ($x$) and a target ($y$), is split into subgroups attached to a knot between an $x$ and an interval in the same $x$ to separate the sub-group. Accordingly, data between any two knots are represented by a piece-wise (i.e., cubic or linear) function whereby basis functions in an adjacent domain intersect at a respective knot. Therefore, the proposed MARS model provides good flexibility in considering the bends, thresholds, and departures from linear built using a matrix of predictors and the predictand [46]. This can capture the non-linear features among all the continuous performance (i.e., online quizzes, assignments, etc.) and the final weighted course grade score datasets.

In terms of the merits of the MARS model built in this study, the regression approach fits a given $x$ data from a subgroup to another subgroup and a spline to another spline, which ensures that adequate data are present in any sub-group. Therefore, the shortest distance between the neighbouring knots is used to avoid any over-fitting of the proposed MARS model. The basis functions, $BF(x)$, are determined from student performance results and later projected on the predictand (i.e., weighted score) matrix [44,45]. Considering an $X$ composed of the vectors ($X_1$, $X_2$...$X_N$), the proposed MARS model is represented as follows

$$\mathbf{Y} = f(\mathbf{X}) + \chi \tag{1}$$

where $N$ = the number of training datum points and $\chi$ = the distribution of errors [46]. The MARS therefore approximates $f(.)$ by applying $BF(x)$ derived from each student performance assessments with a piece-wise function: $\max(0, \mathbf{x} - c)$ with $c$ = the position of a knot [47]. The function $f(\mathbf{X})$ is then constructed as a linear combination of $BF(x)$ and its interactions:

$$f(\mathbf{X}) = \beta_0 + \sum_{n=1}^{n=N} \beta_n BF(\mathbf{X}) \tag{2}$$

In Equation (2), the constant $\beta$ is estimated using least-squares, whereas $f(X)$ is applied as a forward-backward stepwise rule which can identify the knots where the function could vary [43]. At the end of the forward phase, a large MARS model may emerge, and perhaps may also over-fit the training data. We therefore apply a backward phase using generalised cross-validation ($GCV$) regularization to individually delete one or more basis functions. This happens up to a certain point when only the intercept term of the model remains. The $GCV$ (i.e., an estimator for the $N$-training sample's mean square error, $MSE$) [42] is

$$GCV = \frac{MSE}{\left(1 - \frac{enp}{N}\right)^2} \tag{3}$$

In Equation (3), the term $enp$ = the effective number of parameters, with $enp = k + c(k-1)/2$; $k$ = the number of basis functions in the proposed MARS model (incl. the intercept term); $c$ = the penalty (about 2 or 3); and $(k-1)$ = the hinge function knots.

### 2.2. The Benchmark Model 1: Kernel Ridge Regression (KRR)

To benchmark the newly developed MARS model used for the first-year engineering undergraduate mathematics student performance predictions, we now adopt the kernel ridge regression (KRR) method that offers an unlimited non-linear transformation of the predictor features as regressors. Here, the strategy involves kernels and ridge regressions that can avoid model over-fitting issues. The KRR model utilizes regularizations to capture

the non-linear links between all predictors and the respective predictand, and therefore is described mathematically as follows [47,48]

$$\arg min \frac{1}{q} \sum_{0}^{q} ||f_0 - y_0||^2 + \lambda ||f||_H^2 \tag{4}$$

$$f_o = \sum_{p=1}^{q} \alpha_p \omega(x_p, x_o) \tag{5}$$

The Hilbert normed space of Equation (4) is defined as $||.||_H$. For a given $m \times m$ kernel matrix, $K$ is developed by $\omega(x_p, x_o)$ from some fixed predictor variables, where $y$ is the input $q \times 1$ regression vector and is the $q \times 1$ unknown situation vector that reduces as follows [47]:

$$y = (K + \lambda q I) \tag{6}$$

$$\widetilde{y} = \sum_{p=1}^{q} \alpha_o \omega(x_o, \widetilde{x}) \tag{7}$$

In the KRR model training stage, we aim to solve Equation (7) with high accuracy by using linear, polynomial, and Gaussian kernels. For further details on KRR models, the readers can consult several other references, e.g., [47–51].

### 2.3. The Benchmark Model 2: k-Nearest Neighbour (KNN)

The second benchmark model developed for the first-year engineering mathematics undergraduate student performance predictions involves the *k*-nearest neighbour (KNN) technique. The KNN model comprises a supervised machine learning approach for classification and regression problems. As a simple pattern recognition technique, the KNN model is highly effective [52] in modelling a continuous target variable with local non-parametric regressions performed using a function-based approximator [53]. This technique can potentially discover the past data points that are most closely related to the current sequence whilst integrating their future values to estimate the current sequence's next predicted value [54]. The algorithm comprises a matrix $X_t$ with $t = 1, \dots n$ transformed into *d*-dimensional vectors

$$X_t^{d, \ \tau} = (X_t, X_{t-\tau} \dots X_{t-(d-1)\tau}) \tag{8}$$

In Equation (8), $d$ = the number of lags and $\tau$ = the delay parameter. When $\tau$ is assumed as 1, the resulting time series of vectors are:

$$X_t^d = (X_t, X_t, \dots\dots, X_{t-(d-1)}), \ where \ t = d, \dots\dots, n \tag{9}$$

Here, $X_t^d$ is a vector of $d$ consecutive observations in the *d*-dimensional space. The distance between the last vector and each vector in the time series $X_t^d$ where $t = d, \dots n - 1$ is computed and the $k$ vectors nearest to $X_n^d$ are assigned as $X_{T1}^d$, $X_{T2}^d$, $\dots\dots X_{Tk}^d$. Considering the neighbouring vectors, $X_{T1}^d$, $X_{T2}^d$, $\dots\dots X_{Tk}^d$, their subsequent values, $X_{T1+1}^d$, $X_{T2+1}^d$, $\dots\dots X_{Tk+1}^d$ are averaged to obtain the predicted value of $X_{n+1}$. For further details on the KNN model, readers can also consult many other references, e.g., [52–54].

### 2.4. The Benchmark Model 3: Decision Tree (DT)

The MARS model is bench-marked against a decision tree (DT) method that represents a powerful, fast, and easy-to-implement knowledge discovery and data mining technique. A proposed DT model has the capability to determine essential patterns present in relatively complicated datasets [55,56]. Many theorists and practitioners are constantly developing DT-based modeling techniques to improve the process accuracy, efficiency, and cost-effectiveness in terms of scientific and business industries based on its importance in data mining, text mining, information retrieval, machine learning, and pattern identification problems.

In general, a decision tree model represents the division of datasets into branches that result in an inverted decision tree with the root node at the top. The object of analysis is therefore a one-dimensional display that reflects the decision tree interface's root node with mathematical formulation. Giving a set of training vectors $x_i \in R^n$, $i = 1, \ldots . l$ and a label vector $y \in R^l$, a decision tree is generated recursively in order to partition the feature space in such a way that the samples with the same labels or similar target values can be grouped.

Let the data comprised of $N_m$ samples at node $m$ be represented by $Q_m$. For each candidate, we split $\theta = (j, t_m)$ consisting of a feature $j$ and threshold $t_m$, partitioning the data into $Q_m^{left}(\theta)$ and $Q_m^{right}(\theta)$ subsets [55,56].

$$Q_m^{left}(\theta) = (x,y)| \; x_j \; <= \; t_m \tag{10}$$

$$Q_m^{right}(\theta) = Q_m{}_m^{left}(\theta) \tag{11}$$

The quality of a candidate split of node m is then computed using an impurity function or loss function $H()$, which depends on the task being solved.

$$G(Q_m, \theta) = \frac{Q_m^{left}}{N_m} H(Q_m^{left}(\theta)) + \frac{Q_m^{right}}{N_m} H(Q_m^{right}(\theta)) \tag{12}$$

We then select the parameters that minimise the impurity

$$\theta^* = argmin_\theta G(Q_m, \theta) \tag{13}$$

Finally, we recurse the subsets $Q_m^{left}(\theta^*)$ and $Q_m^{right}(\theta^*)$ until the maximum allowable depth is reached, $N_m < min_{samples}$ or $N_m = 1$ or $N_m = 1$. For a detailed theory on DT-based models, readers are encouraged to consult references, such as [55,56].

## 3. Research Context, Project Design, and Model Performance Criteria

### 3.1. Engineering Mathematics Student Performance Data

The proposed MARS (and the comparative KRR, KNN, and DT) models developed to predict student performance in the first-year undergraduate engineering mathematics course consider the case of ENM1500 Introductory Engineering Mathematics that is taught at the University of Southern Queensland in Australia. The course welcomes students entering tertiary studies who are undertaking engineering and surveying programs but they require further skills in problem solving and basic mathematical competencies. The course aims to integrate mathematical concepts by introducing topics such as algebra, functions, graphing, exponential, logarithmic and trigonometric functions, geometry, vectors in two-dimensional spaces, matrices, differentiation, or integration. It develops mathematical thinking, interpreting, and solving authentic engineering problems using mathematical concepts. The course also aims to enable students to communicate mathematical concepts more effectively and express solutions to the engineering problems in a variety of written forms.

Therefore, continuous assessments in ENM1500 comprise two online quizzes, Quiz 1 (*Q*1, 5%) and Quiz 2 (*Q*2, 5%) (marked out of 50, administered in Week 3 and Week 11, respectively); Assignment 1 (*A*1, 15%) and Assignment 2 (*A*2, 15%), marked out of 150 (administered in Week 6 and Week 13, respectively); and an examination (*EX*, marked out of 600, 60%) in Week 15 in a regular teaching semester. Based on continuous assessments spread throughout the semester, students are awarded a grade for their weighted course score (*WS*, 100%). The course was developed as part of a major program update and revision of the previous mathematics syllabus to meet the program accreditation requirements under the Institute of Engineers, Australia (IEAust).

The School of Mathematics, Physics, and Computing in the Faculty of Health, Engineering, and Sciences at the University of Southern Queensland administers ENM1500 as a compulsory part of an Associate Degree of Engineering (ADNG) for Agricultural, Civil,

Computer Systems, Electrical and Electronic, Environmental, Mechanical, and Mining Engineering specializations. In addition, it is a core part of the Bachelor of Construction Management (B. CON) for Civil and Management and Associate Degree in Construction Management.This course is also offered to the Graduate Certificate in Science under the High School and Middle/Primary Teaching Specialization to prepare teachers in engineering or technical subjects. To enter the course, students must have completed Queensland Senior Secondary School Studies Mathematics A (General Mathematics) or have equivalent assumed knowledge, and are advised to undertake an online pre-test on tacit knowledge before commencement. This pre-test informs prospective students on areas that need to be revised to ensure satisfactory progression, including recommendations for further work or an alternative study plan, such as the Tertiary Preparation Program. Therefore, the diversity of any given cohort enrolled in this course provides a rich combination of student learning abilities and learning profiles to build and test the prescribed models to predict $WS$.

Our study considers five consecutive years of student performance data (2015 to 2019, i.e., a pre-COVID period) generated by merging online and on-campus course results held three semesters per year and made available from examiner return sheets that are official results provided to the faculty after a rigorous moderation process prior to grade releases. The modelling data had marks for continuous internal assessments (i.e., two online quizzes, $Q1$ & $Q2$, worth 5% each, and two major written assignments, $A1$ & $A2$, worth 15% each), including a final examination score ($EX$, worth 60%) and a weighted score ($WS$) (i.e., overall mark out of 100%) used to allocate a passing course grade. The content of $Q1$ and $Q2$ had four choices per question that students could possibly select for any given question. For both of the quizzes, there were 15 questions (1 mark each), converted to 50 marks total per Quiz. For the assignments, both $A1$ and $A2$ (marked out of 150) were written assignments with a set of problem-solving tasks for entry-level engineering mathematics applications, as well as basic skill builder tasks. For the examination, there were six long-answer-type application questions (600 marks total) completed over two hour examination period.

An ethics application (#H18REA236) was implemented in accordance with the Australian Code for Responsible Conduct of Research (2018) and the National Statement on Ethical Conduct in Human Research (2007). The research work was purely quantitative with artificial intelligence models that were not aimed at predicting any particular student's performance. It did not draw upon personal information, nor did it disclose any student records such as their name, student identification number, gender, and socioeconomic status. Therefore, based on low risk, an expedited ethical approval was provided with pre-conditions that any form of identification attributes, such as the student names, gender, and personal identifiers, must be removed before processing the student performance data.

While pre-processing the data, incomplete records were deleted entirely (e.g., students who had not submitted assessments for particular items or did not take the exam) to prevent bias in the proposed model. While this led to some loss of student performance data from the original five-year record, the naturally lengthy records enabled us to use a total of 739 complete records of quizzes ($Q1$, $Q2$), assignments ($A1$, $A2$), examination scores ($EX$), and weighted score ($WS$) to ensure negligible effects on the capability of the models to predict a passing or a failing grade. As missing data are a major problem for any machine learning model, this pre-processing data procedure has ensured that any potential bias due to a missing predictor value, for example, a missed assignment or a missed quiz mark for a student, does not cause a loss of predictive features in the overall trained model. This problem could arise from an incomplete record used in the MARS model training phase and, such, it was eliminated by using data records where every assessment data point per student had a corresponding $WS$ value. As this research has used real student performance dataset, there was no suitable method for the recovery of any missing point,; therefore, the row with any missing predictor value was deleted prior to the training of the proposed MARS model.

### 3.2. Model Development Stages

Tables 1 and 2 show the first-year undergraduate engineering mathematics student performance statistics for a five-year period between 2015 and 2019. Each years of data, in their own, were considerably insufficient for model convergence. Therefore, individual years of data were pooled into a global set in order to increase the size of the overall dataset required to fully train, validate, and test the MARS model. Figure 2 investigates the extent of association between the continuous assessments (*Q*1, *Q*2, *A*1, *A*2, and *EX*) and weighted scores (*WS*) using scatter plots and linear regression functions.

**Table 1.** Descriptive statistics of ENM1500 Introductory Engineering Mathematics student performance (2015–2019) used to construct the proposed MARS model with the predictors (inputs) as: *A*1: Assignment 1, *A*2: Assignment 2, *A*3: Assignment 3, *Q*1: Quiz 1, and *Q*2: Quiz 2 with the target. The weighted score (*WS*) represents the overall score used to allocate a course grade. Note that a raw mark for each assessment had a different total with a certain percentage contribution towards the final grade.

| Statistical Property | Predictors | | | | | Target |
| --- | --- | --- | --- | --- | --- | --- |
| | *Q*1/50 5% | *A*1/150 15% | *Q*2/50 5% | *A*2/150 15% | *EX*/600 60% | *WS*/100 100% |
| Mean | 46.6 | 120.5 | 46.3 | 119.9 | 359.1 | 69.3 |
| Median | 50.0 | 127.0 | 50.0 | 126.0 | 360.0 | 70.0 |
| Standard Deviation | 5.5 | 26.0 | 6.7 | 26.4 | 141.1 | 17.3 |
| Minimum | 8.0 | 15.0 | 0.0 | 0.0 | 0.0 | 20.0 |
| Maximum | 50.0 | 150.0 | 50.0 | 150.0 | 600.0 | 100.0 |
| Skewness | −2.7 | −1.2 | −3.4 | −1.3 | −0.2 | −0.2 |
| Flatness | 10.1 | 1.4 | 15.7 | 1.9 | −0.9 | −0.8 |

**Table 2.** Cross-correlation coefficients (*r*) of predictor and target variables and the rank of model inputs based on strength of associations between inputs and the target.

| Predictor versus Target | Assessment in Teaching Week | r-Value | Input Rank |
| --- | --- | --- | --- |
| *Q*1 versus *WS* | 2 | 0.407 | 2 |
| *Q*2 versus *WS* | 10 | 0.606 | 3 |
| *A*1 versus *WS* | 5 | 0.262 | 1 |
| *A*2 versus *WS* | 12 | 0.640 | 4 |
| *EX* versus *WS* | 13 | 0.967 | 5 |

Notably, the extent of the associations between online quizzes, assignments, examination scores, and the final grade differs significantly. A positive correlation between all continuous assessment marks and *WS* is evident although the strength of correlation with *EX* is considerably higher (with $r^2 = 0.9356$) followed by *A*2 ($r^2 = 0.409$), *A*1 ($r^2 = 0.367$), and *Q*1 ($r^2 = 0.164$). For example, the lowest magnitude of a correlation is recorded between *A*1 and *WS*, whereas the highest correlation is evident for *EX* and *WS*, whereas marginal differences exist between the correlation coefficient of *A*2 and *Q*2 analysed against *WS*.

The impact of Quiz 2 on the final grade, as evidenced by the weakest correlation of *Q*2 with *WS*, appears to be the lowest with $r^2 = 0.0685$. Each of the assessment pieces are administered at different times of a 15-week teaching semester; thus, using a diverse set of information to examine the extent of association of each assessment on the weighted score and the estimation of weighted scores resulting from the MARS model can be a useful way to implement effective teaching practices prior to the examination period at the end of the semester. Based on the rank noted in Table 2, the input sequence for the proposed models is designed following the order of increasing the importance of predictors and further testing their importance according to the individual inputs used to predict the weighted scores.

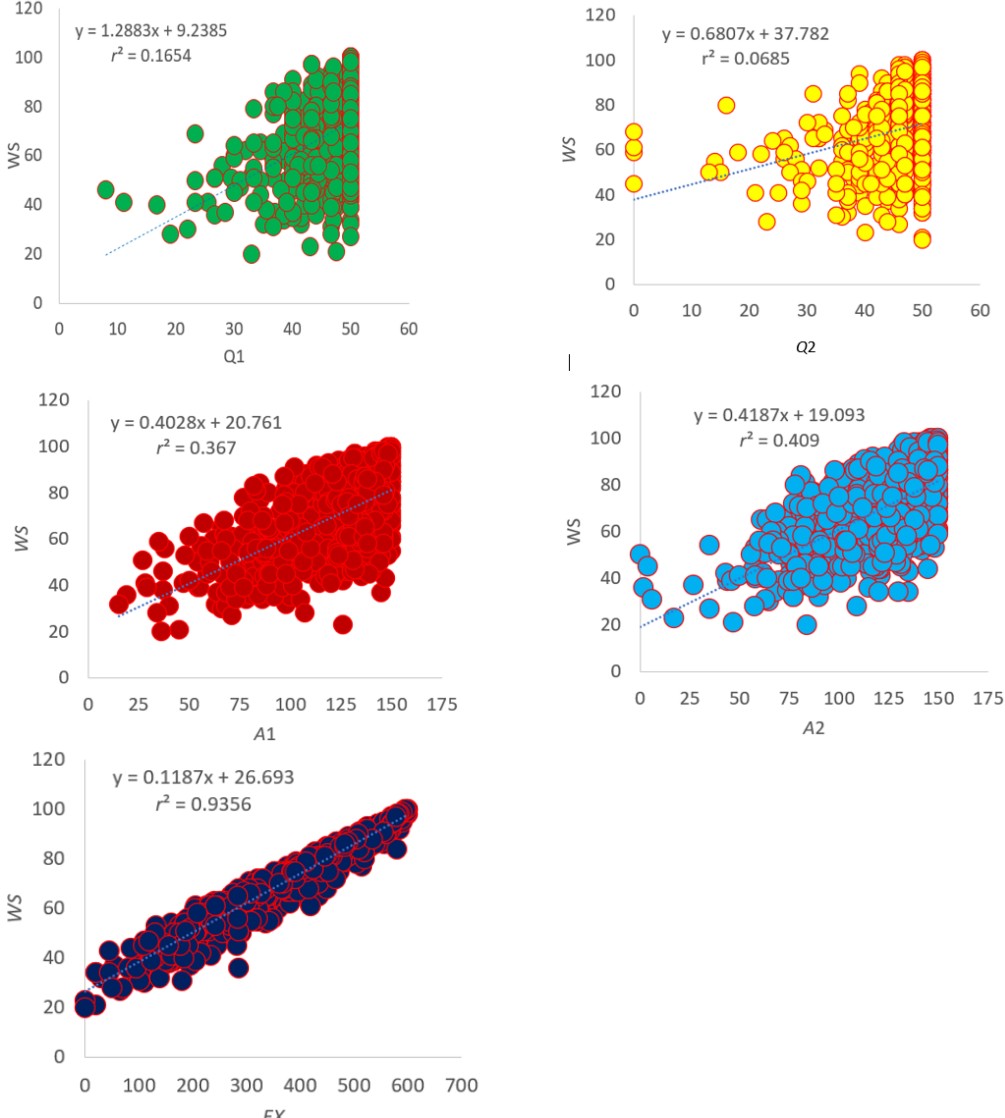

**Figure 2.** Exploring the relationships between each predictor variable and the respective target variable. $Q1$ = Quiz 1; $Q2$ = Quiz 2; $A1$ = Assignment 1; $A2$ = Assignment 2; $EX$ = exam score; $WS$ = weighted score. A least-square regression line with a best fit equation and the coefficient of determination ($r^2$) is shown.

In this paper, two categories of predictive modelling systems are developed. The first is a single-input-based matrix that utilises $A1$, $Q1$, $Q2$, $A3$, and $EX$. These individual models are designated as Models M1–M5. The second category of the modelling system (designated as Models M6–M8) is a multiple-input matrix-based system where the order of the multivariate input combination has been determined statistically. This variable order is selected based on the magnitude of the $r_{cross}$, considering the lowest to the highest level of associations with the target variable ($WS$), as shown in Table 2. This research has built further models by using a combination of the highest correlated input variables. For example, we note that $A2$ and $Q2$ have acquired $r_{cross}$ = 0.606 and 0.640 (i.e., M10), respectively, while adding the relatively low-correlated input $Q1$ and the lowest correlated input $A1$ to further check if the less correlated input variables provide any improvements in the predicted value of $WS$. Table 3 shows the proposed MARS model along with the KNN, KRR, and DT models developed as benchmark methods to comprehensively evaluate the efficacy of the MARS model.

Following this specific modelling strategy, twelve distinct models are built to investigate how the different student assessment datasets impacted the student's overall learning or success in this first-year undergraduate engineering mathematics course taught from Week 1 to 15. To build the proposed MARS (and the benchmark models), all the original data are randomized in the model training phase to ensure greater model credibility for predicting the weighted score. Subsequently, 60% (or 444 rows) of the datasets are allocated to a training set from which 33% (or 145 rows) are selected for model validation purposes. The remainder, 40% (or 295 rows), is used as an independent test set to cross-validate the performance of the proposed MARS and all the other deep learning models.

**Table 3.** Input combinations based on first-year undergraduate engineering mathematics student performance data used to construct the proposed MARS model. Note that Models M1 to M5 are based on single predictor variables, and M6 to M12 are based on multiple predictors used to model the weighted score (*WS*).

| Designated Model | Input Combinations *(Using Predictors in Table 1)* | Data Period S1, S2, S3 | Total Data | Data Points/Period Training (60%) | Validation | Testing (40%) |
|---|---|---|---|---|---|---|
| M1 | $WS = f\{A1\}$ | | | | | |
| M2 | $WS = f\{Q1\}$ | | | | | |
| M3 | $WS = f\{Q2\}$ | | | | | |
| M4 | $WS = f\{A2\}$ | | | | | |
| M5 | $WS = f\{EX\}$ | | | | | |
| M6 | $WS = f\{A1, Q1\}$ | 2015–2019 | 739 records | 444 | 145 (~33%) of training set | 295 |
| M7 | $WS = f\{A1, Q1, Q2\}$ | | | | | |
| M8 | $WS = f\{A1, Q1, Q2, A2\}$ | | | | | |
| M9 | $WS = f\{EX, A2\}$ | | | | | |
| M10 | $WS = f\{EX, A2, Q2\}$ | | | | | |
| M11 | $WS = f\{EX, A2, Q2, Q1\}$ | | | | | |
| M12 | $WS = f\{EX, A2, Q2, Q1, A1\}$ | | | | | |

Table 4 shows the parameter for the MARS, including the KRR, DTR, and KNN models, whereas Table 5 shows the parameters of the most optimal MARS model trained to predict the *WS* using a combination of input variables based on continuous assessments for ENM1500. In accordance with Table 5, we note that the most accurate training performance of the MARS model utilizes 11 basis functions whereby a linear regression model is learned from the outputs of each basis function regressed with the target variable (i.e., *WS*). The basis functions include the intercept term (i.e., 41.719), followed by a product of two or more hinge functions). Consequently, the final prediction is made by summing weighted outputs of all of the basis functions. This step also includes the growing and the generation phase (i.e., the forward-stage) and the pruning or the refining stage (i.e., the backward-stage), as illustrated in Figure 1. This step somewhat resembles the operations of the decision tree (e.g., the DTR) model with each value of each input variable in the training dataset considered as a potential candidate for the basis functions.

The change in the MARS model performance in the backward stage is evaluated using cross-validation of the training dataset (i.e., generalized cross-validation, *GCV*; Table 4). Notably, the optimal model (M12) attained the highest coefficient of determination ($r^2$), the lowest mean square error, and the lowest *GCV*. Moreover, the number of functions is determined automatically, as the pruning process halted when no further improvements were made. Therefore, one benefit of the MARS model was that it only used input variables that improved the performance of the final model, to the extent that the bagging and random forest ensemble algorithms, and the proposed MARS achieved an automatic type of feature selection to generate the most accurate *WS* values in the testing phase.

**Table 4.** The optimal hyperparameter of the proposed (i.e., MARS) and benchmark machine learning models (i.e., DTR, KNN, and KRR)

| Model Name | Hyper-Parameters | Acronym | Optimum |
|---|---|---|---|
| MARS | Maximum degree of terms | max_degree | 1 |
| | Smoothing parameter used to calculate *GCV* | penalty | 3.0 |
| | Regularization strength | alpha | 1.5 |
| | Kernel mapping | kernel | linear |
| KRR | Gamma parameter | gamma | None |
| | Degree of the polynomial kernel | degree | 3 |
| | Zero coefficient for polynomial and sigmoid kernels | coef0 | 1.2 |
| | Maximum depth of the tree | max_depth | None |
| DTR | Minimum number of samples for an internal node | min_sample_split | 2 |
| | Number of features for the best split | max_features | Auto |
| | Number of neighbours | n_neighbors | 5 |
| | Weights | Weights | uniform |
| | The algorithm used to compute the nearest neighbours | algorithm | auto |
| | Leaf-size passed | leaf_size | 25 |
| KNN | Power parameter for the Minkowski metric | p | 2 |
| | The distance metric to use for the tree | metric | minkowski |
| | Additional keyword arguments for the metric | metric_params | none |
| | The number of parallel jobs | n_jobs | int |

### 3.3. Performance Evaluation Criteria

This research adopts visual and descriptive statistics of the observed ($WS_{obs}$) and the predicted weighted scores ($WS_{pred}$) to cross-check the discrepancy of the proposed MARS model using an independent testing dataset not used in the model construction phase. The testing dataset evaluations consider standardised performance metrics to comprehensively evaluate the credibility of the predicted $WS$ in ENM1500 Introductory Engineering Mathematics. The metrics for model evaluation recommended by the American Society for Civil Engineers are root mean square error ($RMSE$), correlation coefficient ($r$), Legate and McCabe's index ($LM$), Nash and Sutcliffe's coefficient ($NSE$), and expanded uncertainty ($U_{95}$) with mathematical representations [57,58].

$$r = \left( \frac{\sum_{i=1}^{N} \left( WS_{pred,i} - \bar{WS}_{Obs,i} \right) \left( WS_{pred,i} - \bar{WS}_{Obs,i} \right)}{\sqrt{\sum_{i=1}^{N} \left( WS_{pred,i} - \bar{WS}_{Obs,i} \right)^2} \sqrt{\sum_{i=1}^{N} \left( WS_{pred,i} - \bar{WS}_{Obs,i} \right)^2}} \right) \tag{14}$$

$$RMSE = \sqrt{\frac{1}{N} \sum_{i=1}^{N} \left( WS_{pred,i} - WS_{Obs,i} \right)^2} \tag{15}$$

$$LM = 1 - \left[ \frac{\sum_{i=1}^{N} |WS_{Obs,i} - WS_{pred,i}|}{\sum_{i=1}^{N} |WS_{Obs,i} - \bar{WS}_{Obs,i}|} \right], 0 \leq LM \leq 1 \tag{16}$$

$$NS = 1 - \left[ \frac{\sum_{i=1}^{N} \left( WS_{Obs,i} - WS_{pred,i} \right)^2}{\sum_{i=1}^{N} \left( WS_{Obs,i} - \bar{WS}_{Obs,i} \right)^2} \right], -\infty \leq NS \leq 1 \tag{17}$$

$$RRMSE = \frac{\sqrt{\frac{1}{N} \sum_{i=1}^{N} \left( WS_{pred,i} - WS_{Obs,i} \right)^2}}{WS_{Obs,i}} \times 100 \tag{18}$$

**Table 5.** Architecture of the proposed MARS model with the basis functions (*BF*), $C_o$ = y-intercept, $y = C_o \pm BF_x$, in terms of the coefficient of determination ($r^2$), the mean square error (*MSE*), and the generalized cross-validation statistic (*GCV*) in the model's training phase.

| Model | MARS Model Equation: $y = C_o \pm BF_x$ | BF | MSE | $R^2$ | GCV |
|---|---|---|---|---|---|
| M1 | $y = 61.98 + 0.5219\ BF_1 - 0.364\ BF_2$<br>$BF_1 = \max(0, x1 - 109);\ BF_2 = \max(0, 109 - x1)$ | 3 | 178.9 | 0.38 | 183.8 |
| M2 | $y = 50.8 + 2.29\ BF_1 + 0.936\ BF_2$<br>$BF_1 = \max(0, x1 - 46);\ BF_2 = \max(0, 40 - x1)$ | 3 | 243.3 | 0.182 | 248.87 |
| M3 | $y = 51.7 + 0.943\ B_1$<br>$BF_1 = \max(0, x1 - 28)$ | 2 | 276.39 | 0.10 | 283.00 |
| M4 | $y = 25.49 + 0.642\ BF_1 - 0.516\ BF_2 + 0.333\ BF_3$<br>$BF_1 = \max(0, x1 - 57.5);\ BF_2 = \max(0, 61 - x1);\ BF_3 = \max(0, 120 - x1)$ | 4 | 167.76 | 0.429 | 173.63 |
| M5 | $y = 48.33 - 0.161\ BF_1 - 0.220\ BF_2 + 0.339\ BF_3$<br>$BF_1 = \max(0, 155 - x1);\ BF_2 = \max(0, x1 - 115);\ BF_3 = \max(0, x1 - 138)$ | 4 | 17.43 | 0.939 | 18.50 |
| M6 | $y = 72.45 + 1.878\ BF_1 - 0.531\ BF_2 - 0.822 BF_3 - 0.346\ BF_4$<br>$BF_1 = \max(0, x2 - 47);\ BF_2 = \max(0, 47 - x2);\ BF_3 = \max(0, x1 - 139);\ BF_4 = \max(0, 139 - x1)$ | 5 | 162.2 | 0.442 | 169.78 |
| M7 | $y = 71.48 + 2.777\ BF_1 + 307.38\ BF_2 - 0.5777\ BF_3 - 3.348\ BF_4 - 3.313\ BF_5 + 2.196\ BF_6 - 0.054\ BF_7 - 2.122\ BF_8\ 0.0757\ BF_9$<br>$BF_1 = \max(0, x1 - 144);\ BF_2 = \max(0, x2 - 47);\ BF_3 = \max(0, 47 - x2);\ BF_4 = BF_2\ \max(0, 149 - x1);\ BF_5 = BF_2\ \max(0, x1 - 57);$<br>$BF_6 = \max(0, 36 - x3);\ BF_7 = \max(0, x3 - 36)\ \max(0, 122 - x1);\ BF_8 = \max(0, 43 - x3);\ BF_9 = \max(0, x3 - 43)\ \max(0, 101 - x1);$ | 10 | 154.144 | 0.442 | 169.90 |
| M8 | $y = 72.62 + 0.645\ BF_1\ 0.267\ BF_2 + 2.209\ BF_3 - 3.928\ BF_4 - 0.345\ BF_5 + 0.002\ BF_6 - 0.313\ BF_7 + 1.187\ BF_8$<br>$BF_1 = \max(0, x4 - 33);\ BF_2 = \max(0, 33 - x4);\ BF_3 = \max(0, x1 - 47);\ BF_4 = BF_3\ \max(0, x2 - 149);\ BF_5 = \max(0, 137 - x2);$<br>$BF_6 = BF_5\ \max(0, 146 - x4);\ BF_7 = \max(0, x2 - 137)\ \max(0, x3 - 47);\ BF_8 = \max(0, x2 - 145);$ | 9 | 124.145 | 0.547 | 137.82 |
| M9 | $y = 46.838 + 0.105\ BF_1 - 0.133\ BF_2 + 0.151\ BF_3 - 0.152\ BF_4 + 0.002\ BF_5 + 0.001\ BF_6$<br>$BF_1 = \max(0, x2 - 205);\ BF_2 = \max(0, 205 - x2);\ BF_3 = \max(0, x1 - 77);\ BF_4 = \max(0, 77 - x1);\ BF_5 = BF_2\ \max(0, x1 - 109);$<br>$BF_6 = BF_2\ \max(0, 109 - x1);$ | 7 | 5.081 | 0.982 | 5.60 |

**Table 5.** *Cont.*

| Model | MARS Model Equation: $y = C_o \pm BF_x$ | BF | *MSE* | $R^2$ | GCV |
|---|---|---|---|---|---|
| M10 | $y = 39.665 + 0.103\ BF_1 + 2.375\ BF_2 + 0.001\ BF_3 - 0.013\ BF_4 - 0.015\ BF_5 + 0.004\ BF_6 + 0.016\ BF_7 - 1.307\ BF_8 - 0.009\ BF_9 - 0.018\ BF_{10}$ $+ 1.427\ BF_{11} - 2.465\ BF_{12} + 0.010\ BF_{13}$ $BF_1 = \max(0, x3 - 205);\ BF_2 = \max(0, 77 - x2);\ BF_3 = \max(0, 205 - x3)\max(0, x2 - 115);\ BF_4 = \max(0, 44 - x1)\max(0, x2 - 57.5);$ $BF_5 = \max(0, 44 - x1)\max(0, 57.5 - x2)\ ;\ BF_6 = \max(0, x2 - 77)\max(0, x1 - 44);\ BF_7 = \max(0, x2 - 77)\max(0, 44 - x1);\ BF_8 = \max(0, x2 - 80);$ $BF_9 = \max(0, 205 - x3)\max(0, x1 - 30);\ BF_{10} = \max(0, 205 - x3)\max(0, 30 - x1);\ BF_{11} = \max(0, x2 - 74);\ BF_{12} = \max(0, 74 - x2);$ $BF_{13} = \max(0, x1 - 44)\max(0, 210 - x1)$ | 14 | 4.45 | 0.985 | 3.565 |
| M11 | $y = 45.628 + 0.102\ BF_1 - 0.115\ BF_2 + 0.494\ BF_3 - 0.259\ BF_4 + 0.106\ BF_5 - 0.042\ BF_6 + 0.003\ B_7 + 0.006\ B_8 + 0.007\ BF_9 - 0.005\ BF_{10} - 0.356\ BF_{11}$ $+ 0.063\ BF_{12} - 0.015\ BF_{13}$ $BF_1 = \max(0, x4 - 200);\ BF_2 = \max(0, 200 - x4);\ BF_3 = \max(0, x3 - 77);\ BF_4 = \max(0, 44 - x2);\ BF_5 = BF_2\max(0, x1 - 44)\max(0, x1 - 47.5);$ $BF_6 = \max(0, 77 - x3)\max(0, x1 - 43);\ BF_7 = BF_4\max(0, x3 - 106);\ BF_8 = BF_4\max(0, 106 - x3);\ BF_9 = BF_2\max(0, 42 - x2);$ $BF_{10} = BF_2\max(0, 37 - x1);\ BF_{11} = \max(0, x3 - 81);\ BF_{12} = \max(0, 81 - x3)\max(0, x1 - 46.67);\ BF_{13} = \max(0, 81 - x3)\max(0, 46.67 - x1);$ | 14 | 3.187 | 0.986 | 4.11 |
| M12 | $y = 41.719 + 0.0999\ BF_1 - 0.1000\ BF_2 + 0.101\ BF_3 - 0.0999\ BF_4 + 0.100\ BF_5 - 0.102\ B_6 + 0.0987\ BF_7 - 0.0978\ BF_8 + 0.0989\ BF_9 - 0.0938\ BF_{10}$ $BF_1 = \max(0, x5 - 200);\ BF_2 = \max(0, 200 - x5);\ BF_3 = \max(0, x4 - 77);\ BF_4 = \max(0, 77 - x4);\ BF_5 = BF_2\max(0, x2 - 80);$ $BF_6 = \max(0, 80 - x2);\ BF_7 = BF_4\max(0, x3 - 26);\ BF_8 = BF_4\max(0, 26 - x3);\ BF_9 = \max(0, x1 - 33.33);\ BF_{10} = \max(0, 33.33 - x1);$ $BF_{11} = \max(0, x3 - 81);\ BF_{12} = \max(0, 81 - x3)\max(0, x1 - 46.67);\ BF_{13} = \max(0, 81 - x3)\max(0, 46.67 - x1);$ | 11 | 0.079 | 0.997 | 0.0902 |

$$u_{95} = 1.96\sqrt{SD^2 + RMSD^2} \tag{19}$$

where $RMSD = \frac{100}{WS_{Obs,i}}\sqrt{\frac{(WS_{pred,i} - WS_{Obs,i})^2}{N}}$.

Note that $WS_{Obs}$ and $WS_{Pred}$ are the observed and predicted $i^{th}$ values of the WS; $\overline{WS}_{Obs,i}$ and $WS_{pred,i}$ are the observed and predicted $WS$ in the testing phase; and $N$ = the number of data points.

## 4. Results and Discussion

The results generated by the newly proposed MARS (and the comparative models) are presented with respect to their predictive skills in emulating the weighted course score used to allocate a final grade in ENM1500 Introductory Engineering Mathematics.

Table 6 compares the observed and the predicted $WS$, presented in terms of the correlation ($r$) and root mean square error ($RMSE$) for diverse input combinations (i.e., M1 to M12). It becomes immediately apparent that the MARS model, designated as M5 with $WS = f\{EX\}$ and M12 ($WS = f\{EX, A2, Q2, Q1, A1\}$), is the most accurate model compared with M1 to M12. However, the performance of the MARS model designated as M5 and M12 input combination appears to far exceed the performance of DTR, KNN, and KRR models in terms of the tested error and the correlation between the observed and the predicted weighted score.

**Table 6.** Root mean square error ($RMSE$) and correlation coefficient ($r$) between observed $WS$ and predicted $WS$ generated by the proposed MARS model compared with three different benchmark (i.e., DTR, KNN, KRR) models.

| Designated Model | Predicted Error: *RMSE* | | | | Correlation Coefficient (*r*) | | | |
|---|---|---|---|---|---|---|---|---|
| | MARS | DTR | KNN | KRR | MARS | DTR | KNN | KRR |
| M01 | 14.26 | 16.06 | 15.74 | 14.30 | 0.574 | 0.472 | 0.452 | 0.568 |
| M02 | 16.07 | 16.37 | 15.88 | 16.01 | 0.401 | 0.373 | 0.438 | 0.408 |
| M03 | 16.93 | 17.21 | 17.66 | 16.81 | 0.269 | 0.222 | 0.184 | 0.285 |
| M04 | 13.81 | 14.96 | 14.52 | 13.75 | 0.622 | 0.524 | 0.556 | 0.628 |
| **M05** | **5.76** | **6.54** | **5.95** | **5.89** | **0.963** | **0.950** | **0.961** | **0.960** |
| M06 | 13.69 | 16.79 | 14.55 | 13.80 | 0.620 | 0.478 | 0.580 | 0.607 |
| M07 | 13.69 | 16.73 | 14.32 | 13.77 | 0.620 | 0.496 | 0.597 | 0.608 |
| M08 | 12.64 | 15.95 | 13.28 | 12.66 | 0.688 | 0.536 | 0.655 | 0.686 |
| M09 | 4.58 | 5.14 | 4.75 | 4.67 | 0.986 | 0.978 | 0.985 | 0.985 |
| M10 | 4.30 | 5.24 | 4.79 | 4.66 | 0.990 | 0.978 | 0.986 | 0.988 |
| M11 | 4.21 | 5.05 | 5.21 | 4.64 | 0.991 | 0.978 | 0.984 | 0.990 |
| **M12** | **3.29** | **4.39** | **4.60** | **3.89** | **0.998** | **0.987** | **0.990** | **0.994** |

Among the input combinations for models designated as M5 and M12, we also noticed that the model M12 used in the MARS model yields ≈ a 42% lower error, whereas that for the DTR model is 32.9% lower, KNN is 22.7% lower, and KRR is 33.9% lower than M5. This shows that the input combinations used in case of M12 with the variable $EX$, $A2$, $Q2$, $Q1$, and $A1$ can improve the prediction of $WS$ compared with the $EX$ as a single input.

In a physical sense, this means that the influence of online quizzes and written assignments on ENM1500 student outcomes is significant. However, it is imperative to note that for the optimal input combination, the proposed MARS model far exceeds the performance of the DTR, KRR, and KNN models, as measured by the errors attained in their testing phase. This result concurs with the initial correlation coefficients stated in Table 6, where the highest degree of agreement between the observed and tested $WS$ is evident by the largest $r$ value for the case of the MARS model relative to the counterpart models.

Interestingly, we note that among the combination of single predictors, the proposed MARS model (M5) with $EX$ as input registers the best performance ($r = 0.963$; $RMSE = 5.76$),

which is followed by the model with *A*2 (M4) and *A*1 (M1). Notwithstanding this, the worst performance is registered for the case of M3 (with *Q*2 as input). This indicates that Quiz 2 has the weakest influence on the weighted score while the examination score has the strongest influence on the weighted score (or final grade for ENM1500 students).

It is noteworthy that a diverse range of model combinations prepared by adding the predictor variables in an ascending order (i.e., M9, M10, M11, M12) reveals a significant improvement in the accuracy of the tested dataset by a margin of ≈ 20% to a 43% reduction in the predicted RMSE values. Similarly, in terms of the *r* values, the improvement for the case of the MARS model is ≈ 2% to 4%, as we analyzed models M9 to M12, respectively. If we only compared the results for the input combination case M12 where all of the predictor variables (i.e., *EX*, *A*2, *Q*2, *Q*1, and *A*1) are used, the proposed MARS model generates the best performance (i.e., *r* = 0.998; *RMSE* = 3.29 followed by KRR with *r* = 0.994; *RMSE* = 3.89, DTR with *r* = 0.987; *RMSE* = 4.39 and KNN with *r* = 0.990; *RMSE* = 4.60). By contrast, the model for the input combinations prepared in a descending order, namely M6, M7, and M8, yielded a comparatively poor performance.

To appraise the proposed MARS model, we now show in Figure 3 the Nash–Sutcliffe's coefficients (Equation (17)) employed to assess the predictive skills of all models. As the *NSE* is calculated as one minus the ratio of the error variance of the modelled data divided by the variance of the observed data, a perfect model with an estimation error variance equal to zero is expected to record the Nash–Sutcliffe efficiency of unity, whereas the model that produces an estimation error variance equal to the variance of the observed data will produce a trivial value of *NSE*. Therefore, for a model with NSE close to zero, it would have the same predictive skill as the mean of the data in terms of the sum of the squared error.

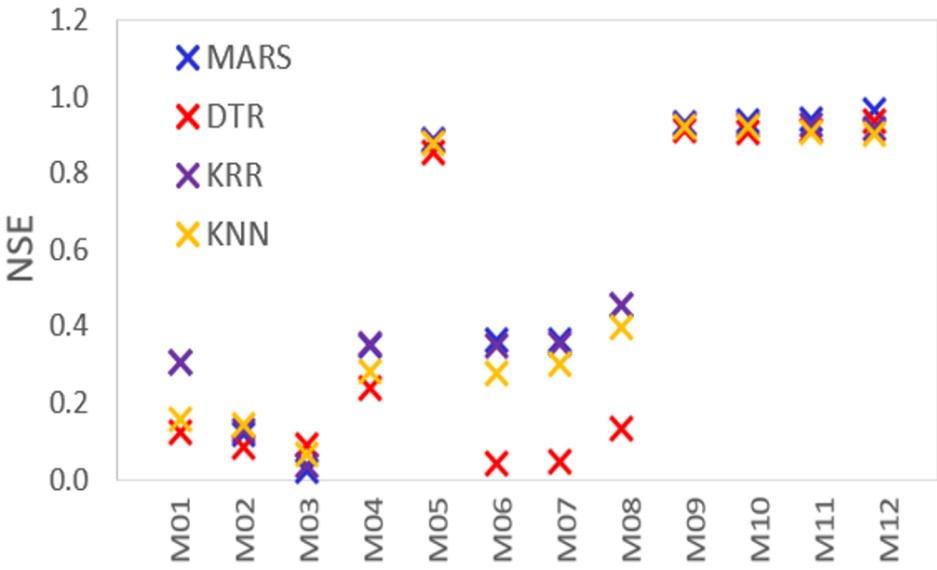

**Figure 3.** Comparative analysis of machine learning methods (i.e., MARS, vs. KNN, KRR, and DTR) employing Nash and Sutcliffe's coefficient (*NSE*) computed between the predicted *WS* and the observed *WS* in the testing phase.

Figure 3 shows that MARS model designated as M9 to M12 yielded *NSE* value close to unity, although M12 (with $WS = f\{EX, A_2, Q_2, Q_1, A_1\}$) appears to be a better fit compared with M9, M10, and M11 where *EX*, *Q*1, and *A*2 have been excluded from the predictor variable list. It is also of note that DTR, KRR, and KNN are relatively less accurate than MARS to ascertain its superior skill in predicting the weighted scores for ENM1500 students. Interestingly, all the four algorithms with input combinations designated as M5 produce quite an accurate simulation of *WS*, which concurs with Table 6, where the same model yielded a significantly high correlation (*r* = 0.950–0.963) and a relatively low *RMSE* (5.76–5.89). Taken together with *RMSE* and *r* values, the high *NSE*

for M5 shows that the examination score remains the most significant predictor of weighted scores. However, the importance of online quizzes and written assignments remain non-negligible (see designated inputs and results for M9–M11).

Figure 4 compares the percentage change in the root mean square error generated by the proposed MARS model (vs. DTR, KNN & KRR) for input combinations M1–M12. The purpose is to evaluate the exact level of improvement attained by the MARS model against that of the comparative counterpart models. Interestingly, the most significant improvement in the proposed MARS model performance is attained for the input combination M12, where it records a significant performance edge over KNN (30% improvement), followed by DTR ($\approx$26% improvement) and KRR (15% improvement). It is quite interesting to note that model M5 (which attains a relatively high *NSE* and a relatively low *RMSE*: Figure 3; Table 6) did not reveal a large improvement in terms of percentage change in *RMSE* values. This seems to suggest that although *EX* is highly correlated with *WS* (Figures 2 and 3), the inclusion of other assessment marks, such as online quizzes and written assignments, leads to a dramatic improvement in the MARS model's ability to predict the weighted scores accurately. This suggests that the influence of continuous assessment remains quite significant on the final grade of a majority of the ENM1500 students.

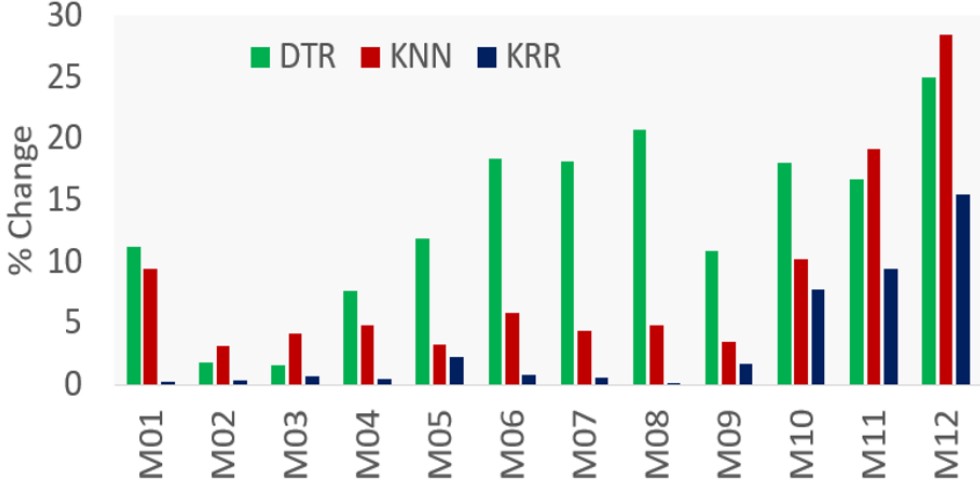

**Figure 4.** Change in the predicted value of the root mean square error (*RMSE*) deduced by comparing the *RSME* for the proposed MARS model relative to the *RSME* generated by the benchmark (i.e., DTR, KNN, and KRR) model. Note:% Change = $| (RMSE_{MARS} - RMSE_{DTR,KNN,KRR})/RMSE_{MARS} | \times 100$.

We now compare the capability of the proposed MARS model used to predict the weighted score using an expanded uncertainty ($U_{95}$) metric calculated by multiplying the combined uncertainty with a coverage factor ($k$ = 1.96 that is used for an infinite degree of freedom) to represent the errors at the 95% confidence level. In addition, the Legate and McCabe index (*LM*), to more stringent metric than the *NSE*, is also used to benchmark the proposed MARS model against the comparative models for a range of input combinations (i.e., M1–M12). As shown in Figure 5 for the testing phase, the magnitude of $U_{95}$ and *LM* are in tandem with each other, whereby the lowest value of $U_{95}$ and the highest value of *LM* are attained by the proposed MARS model, particularly for the case of Model M12.

Figure 6 plots a Taylor diagram where the root-mean-square-centred difference (*RMSD*) and the standard deviation are considered against the correlation coefficient of observed and predicted weighted score of the ENM1500 students. In this case, we plotted the objective model (i.e., MARS) in a separate panel compared with KNN, KRR, and DTR models for the complete set of input combinations M1–M12. There appears to be a clear separation of the results for M9-M12 from that of the other designated inputs for all four types of models. However, the MARS model (for M12) outperforms all counterpart models for this input combination to ascertain its outstanding ability to predict weighted scores.

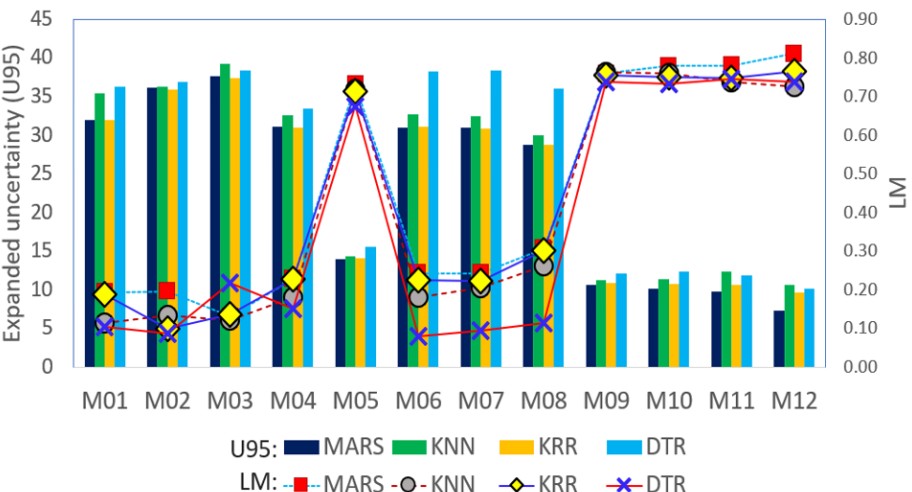

**Figure 5.** Evaluation of the predictive skill of all machine learning models with various input combinations developed to predict the weighted score, shown in terms of expanded uncertainty ($U_{95}$) and the Legates and McCabe index (*LM*) in the testing phase. Note that the proposed MARS model attains the highest value of *LM* and the lowest value of $U_{95}$.

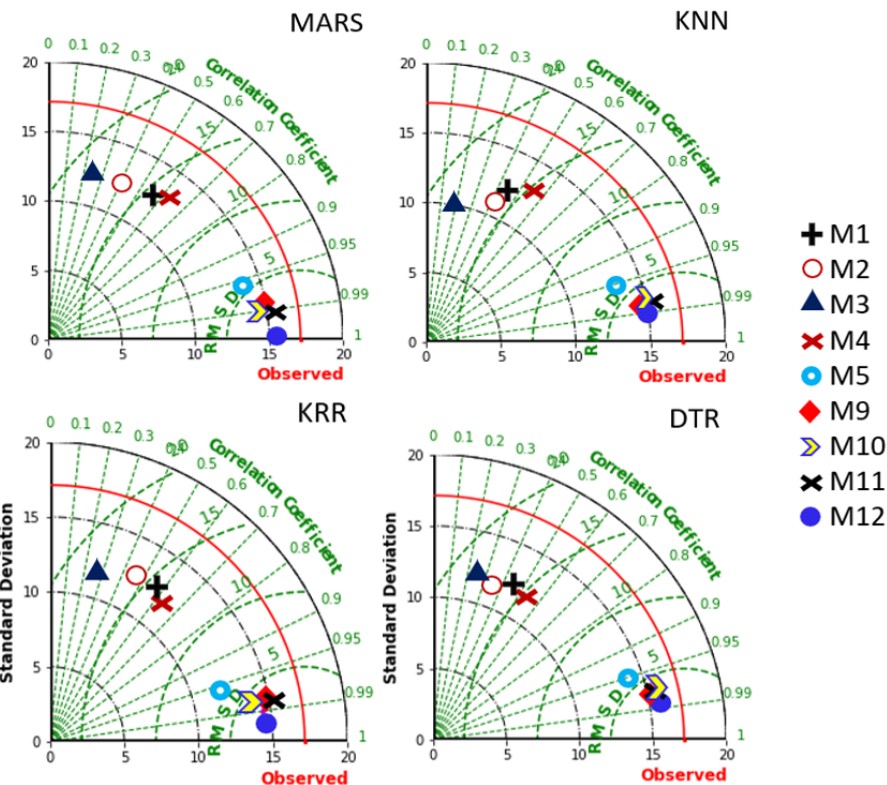

**Figure 6.** Taylor diagram showing the correlation coefficient between the predicted and the observed weighted scores, including the standard deviation and root mean square centred difference for the machine learning models (i.e., MARS, KNN, KRR, and DTR) and including different feature (or input) combinations M1–M5, and M9–M12.

Figures 7a,b and 8a,b represent a scatterplot of the predicted and the observed *WS* for the proposed MARS model and the other comparative models. According to the scatterplot, the coefficient of determination ($r^2$) is associated with the goodness-of-fit between predicted and observed *WS* as well as a line of least-square fit with appropriate equation $y = mx + c$, where "*m*" = the gradient and "*c*" = the regression line y-intercept. The proposed model

with all predictors (i.e., M12) significantly outperformed the baseline models and all other input combinations in terms of the highest $r^2$ value.

When the magnitudes of these parameters are stated in pairs ($m|r^2$), the proposed MARS model with M12 reports the values closest to unity at 0.998 | 0.907 ($m|r^2$), followed by KRR for M12 (0.993 | 0.845). Additionally, the MARS model also showed a subsequent improvement measured by the single predictor variable-based model to all the predictor-based models (i.e., M12), signifying the contributions of all student evaluation components in assessing the student-graded performance. Therefore, the proposed MARS model with M12 input combination can be said to be well suited for predicting the weighted scores of ENM1500 students.

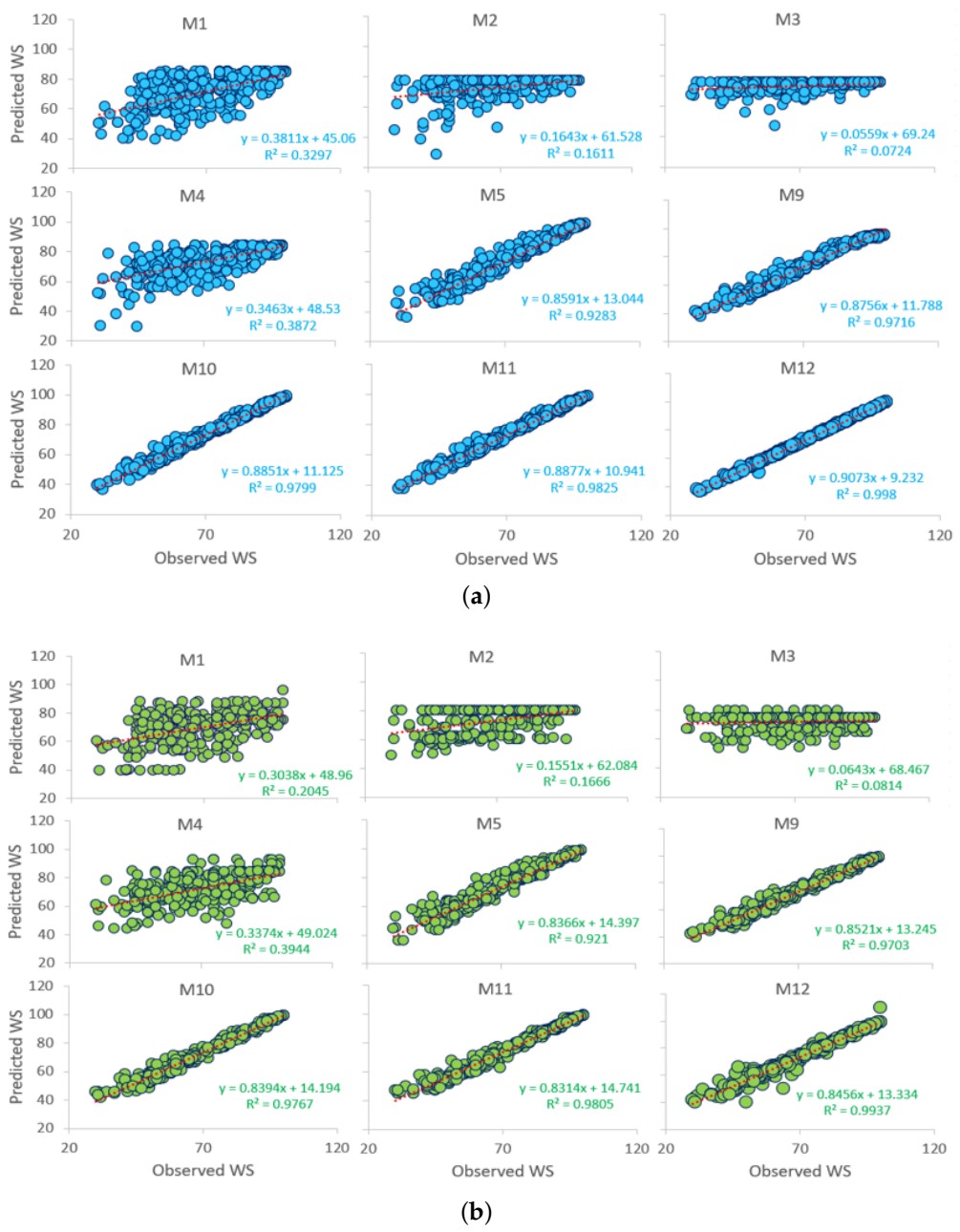

**Figure 7.** Scatter plot of the predicted weighted score (*WS*) versus the observed *WS* in the testing phase in terms of the nine different sets of feature (input) combinations used to predict *WS*. Least-square regression line $y = mx + C$ and the coefficient of determination ($r^2$) are shown in each sub-panel. (**a**) MARS, (**b**) KNN.

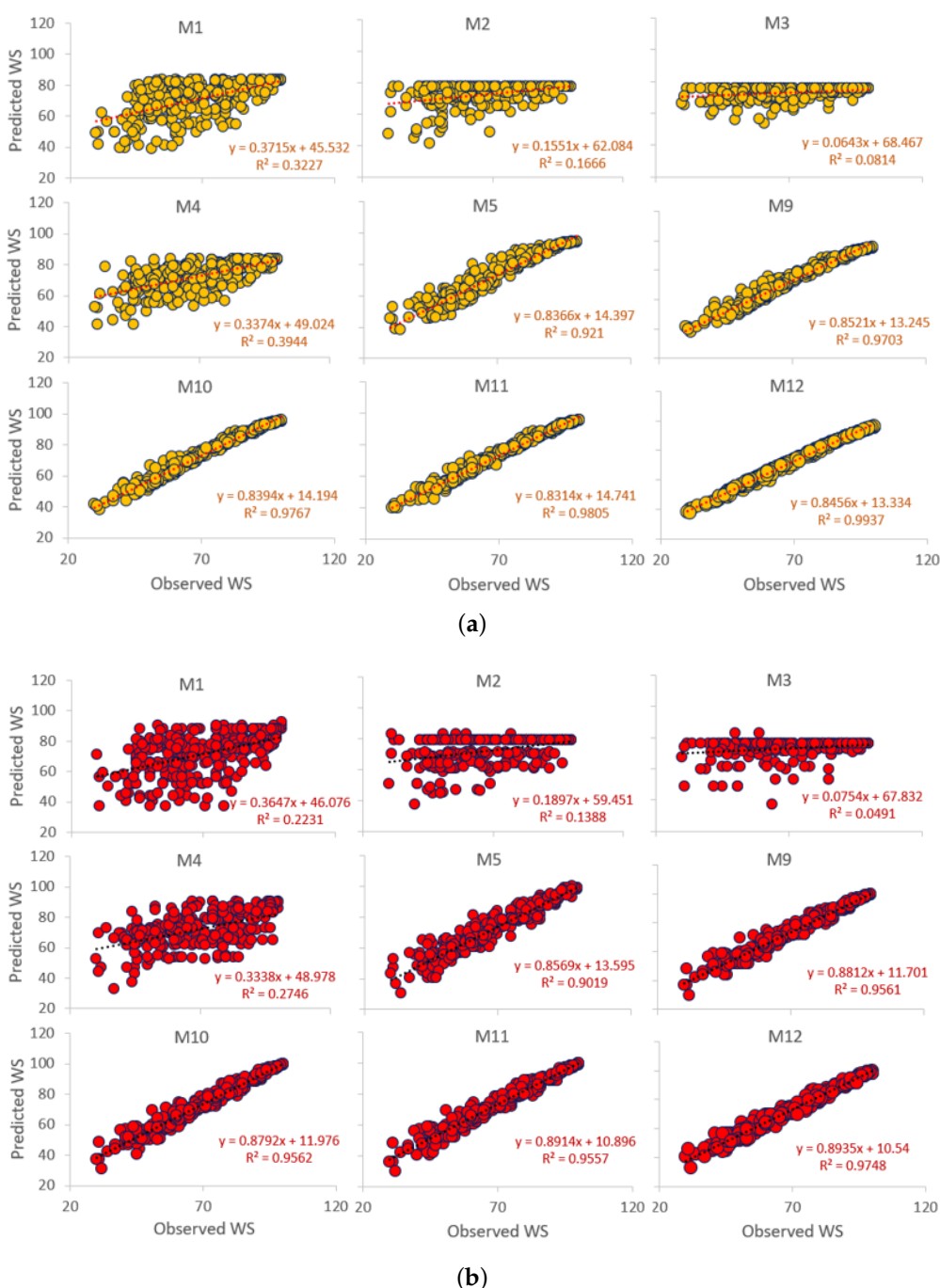

**Figure 8.** Caption identical to Figure 7 except for (**a**) KRR and (**b**) DTR.

We now show the discrepancy ratio ($DR$) as a metric to examine the robustness of all the developed models in Figure 9. Note that the $DR$ metric indicates whether a model over- or under-estimates the value of a weighted score so that a DR value close to unity is expected to indicate a predicted value closely resembling the observed value. Notably, across the tested data points, the proposed MARS model (with M12 as the input combination) attained 90% and 98% of the observations distributed within the $\pm 10\%$ and $\pm 30\%$ band error, respectively. For the other input combinations, the outliers are somewhat higher, which indicated a poor prediction by the MARS model.

Further evaluation of the proposed MARS model is accomplished by investigating the empirical cumulative distribution function ($ECDF$). Here, we show the absolute predicted error ($|PE|$) for the case of Model M12 in Figure 10. The figure demonstrates that about

95 percent of all $|PE|$ values generated by the MARS model fall within the $\pm5.60$ error bracket, followed by $\pm6.71$ for the KRR, $\pm7.94$ for the DTR, and $\pm8.30$ for the KNN model, respectively. The mean value of the predicted error for the proposed MARS is $\approx2.824$ (vs. 3.290–3.553 for KRR, DTR, and KNN models), whereas the standard deviation is $\approx1.688$ for MARS (vs. 2.078–2.920) for a total of 295 tested values of the weighted score in an independent testing phase. Taken together, Figures 9 and 10 demonstrate the efficacy of the proposed MARS model to generate relatively accurate weighted scores for ENM1500 students.

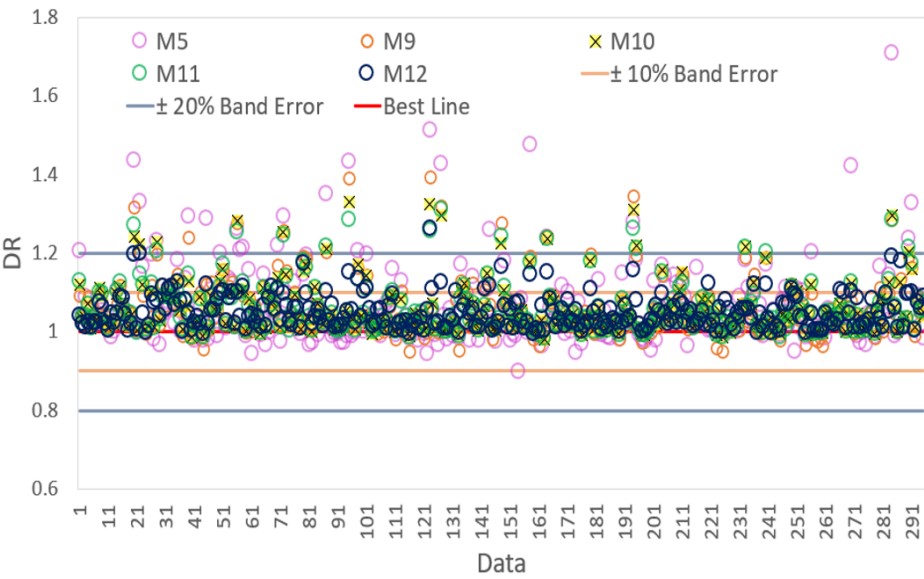

**Figure 9.** Discrepancy ratio, *DR* (i.e., the predicted *WS* divided by the observed *WS*), for the proposed MARS model within the $\pm10\%$ and $\pm20\%$ error bands for all tested data points.

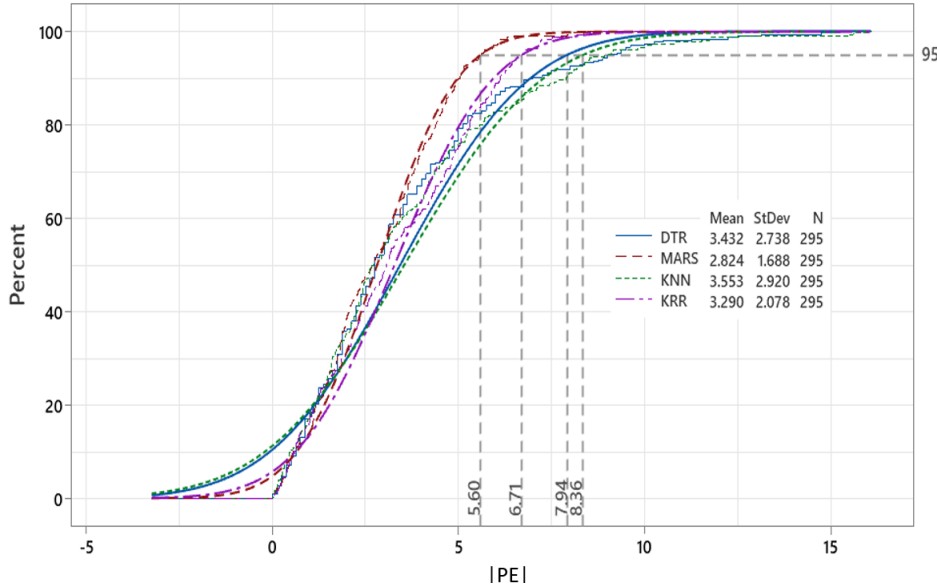

**Figure 10.** Empirical cumulative distribution function (*CDF*) showing the predicted error $|PE|$ for the MARS, versus DTR, KNN, and KRR models for the model denoted as M12. Note that the MARS model converges more rapidly for $|[PE] > 2.5$, compared to the benchmark models.

## 5. Further Discussion, Limitations of This Work, and Future Research Direction

In this research, the performance of a novel MARS model was shown to far exceed three machine learning models for the specific case of Introductory Engineering Mathematics (ENM1500) taught at the University of Southern Queensland, Australia. A statistical and

visual comparison of observed and predicted weighted scores used to determine final grades showed different levels of association of continuous assessments—evaluated both as single predictors and a combination of predictors based on the correlation coefficient of each assessment item (see Table 3, Table 6). Although the examination score was the most significant indicator of success in the course in terms of statistical evaluations (Figure 2) and results (Figures 3–10, the inclusion of online quizzes and written assignments led to a dramatic improvement in the predicted accuracy of the final course grade. This outcome highlights the critical role of every assessment in producing a successful grade. The effect of each input combination (and its contributory role in leading to a successful grade) was also notable, suggesting that the MARS model could be a useful stratagem for engineering mathematics educators in developing early intervention programs to redefine their teaching strategies as a semester progresses. As each assessment was spread throughout the 15-week teaching semester, the application of the MARS model fed with each assessment mark early into the semester could be a useful tool to develop scenarios of student success or failure.

Despite the superior performance of the MARS model, there are still some limitations that warrant further investigation. In this study, the dataset used to train the model was not partitioned into factors that could further limit the model's generality. For example, student performance data divided into gender-based, socio-economic status, pre-requisite knowledge of mathematics, student marks based on personality traits, learning styles, and psychological well-being could also be considered to develop separate models for each type of student cohort. For example, Fariba [24] found correlations between such factors that lead learners to a higher level of learning and redefine their self-satisfaction and enjoyment during their learning journeys. To investigate such factors, a much larger dataset from more than one academic institution with comparative specifications of their engineering mathematics courses could provide support for developing a more robust generic model for customised predictions of undergraduate student success at different institutions. Furthermore, in this study, we build the MARS model by pooling all years of data together to create a universally diverse dataset with a relatively lengthy record (see Table 1). While this approach ensured the MARS model has enough data for the training, validating, and testing stages, and that each years of data, in their own, were considerably insufficient for model convergence, our study does have limitations in terms of not considering individual years of data to train each year-by-year model. A further study utilizing each years of data, or groups of years of data to further train the MARS model, is warranted to check for any model discrepancies.

In a future study, one could categorize datasets into different performance thresholds (or grades) to develop classification models that investigate the relatively poor-performing students to be identified early, thereby allowing the educational institution's management to intervene and improve their performance [59]. Unfortunately, it is difficult to scale the existing single classifier-based predictive models from one context to another or to attain a general model across a diverse range of learners. Therefore, a classification model studying the categorized dataset of low to moderate performers could be developed. In this study, datasets from only one university were used, so a predictive model constructed for one course at one institution may not apply more generally to another method or another institution. Therefore, the concept of integrated multiple classifiers for datasets from various universities and courses may lead to more robust and tailored strategies to predict students' academic success. The idea behind combining such datasets through various classifiers is that the different classifiers, each of which are expected to use a distinct data representation, concept, and modelling technique, are more likely to produce classification performances with varying generalization patterns [60] that can lead to a more universal model. Some scholars have demonstrated that the approach using multiple classifier models that aim to minimize the classification error while maximizing the generalization skills of the model [61]. Therefore, in future applications, the MARS model could be made flexible, generalizable, and scalable through predictive modelling datasets using multiple classifier systems.

Different factors influence students' academic performance, such as socioeconomic status, family atmosphere, schooling history and available training facilities, relational networks of the persons, and student–teacher interactions. These factors are parts of the academic problems that cannot be resolved without addressing the essential aspect. However, these factors may sometimes contribute less to some of the poor performance and academic problems observed among students but can be attributed to the poor performance at the psychological organization level, i.e., motivational and personality factors. In the face of severe external resources limitation, such as socioeconomic constraints, as seen in many rural areas, schools must rely on other resources to ensure they achieve their goals. Although some students in rural schools may have resources to support positive academic outcomes at home, most of them may be facing problems of resource availability and other family-related issues such as single parenting, low socioeconomic status, low parental education, etc., which may lead to low performance and risk of dropout. These issues could be the subject of further investigation to extend the use of MARS or other approaches to a more diverse set of targeted outcomes.

## 6. Conclusions

Predicting student performance is a crucial skill for educators, not only for those striving to provide their students with the opportunity to be productive in their fields of study but also for those educators who need to manage the teaching and learning resources required to deliver a quality education experience. In this study, the undergraduate Introductory Engineering Mathematics student weighted scores were predicted successfully using continuous assessment marks by developing a new multivariate adaptive regression splines (MARS) model using specific datasets from the University of Southern Queensland, Australia.

The model was constructed using ENM1500 (Introductory Engineering Mathematics) data over five years from the University of Southern Queensland, Australia, to simulate the overall student marks leading to a grade using online quizzes ($Q1$ & $Q2$), written assignments ($A1$ & $A2$), and the final examination score ($EX$). The model simulations showed that the examination, assignments, and quizzes together could be used to model the weighted score, although there was a significant influence of each assessment on the weighted score. Based on statistical and visual analysis of predicted and real weighted scores, a MARS model captured the dependence structure between the predictor and the target variable. Compared with a decision tree regression (DTR), kernel ridge regression (KRR), and $k$-nearest neighbour (KNN) model, the MARS model was able to capture the interaction between variables perfectly as an efficient and fast algorithm during computation and was very robust to the outliers in the weighted score. The MARS model registered the lowest predicted root mean square error ($RMSE$) 5.76% vs. 5.89–6.54% for the three benchmark models, attaining the highest correlation of $\approx 0.963$ vs. 0.950–0.961. With assignments and quiz marks added to the input list, the MARS model accuracy improved significantly, yielding a lower $RMSE$ (3.29%) and a larger correlation of 0.998 for predicted vs. observed $WS$. This demonstrated the usefulness of the model to educators. In particular, the models developed can assist the educators in demonstrating how future students learning needs in terms of, or evidenced by, continuous assessments such as assignments may impact their examination performance. The predicted student marks in these assessments can help educators to reflect on their teaching strategies, or to identify deficits in teaching methods, their effectiveness, and student's unique learning styles for a more productive planning and early intervention to prevent failures. The results confirmed that the proposed MARS model was superior to four other benchmark models, as demonstrated by the lowest expanded uncertainty and the highest Legates–McCabe index, with a Taylor diagram and empirical error plots for comparing predicted and observed weighted score. Therefore, such models can be used as an early intervention tool by using early assessments (e.g., quizzes or assignments) to predict either examination outcomes or final grades.

We conclude that this study used only students' quizzes, examination, and assignment results to construct machine learning regression models, and therefore has ignored some of the other personal variables that may influence student outcomes. These variables are socioeconomic status, family atmosphere, schooling history and available training facilities, relational networks, student–teacher interactions, and many others. While the study has aimed to develop a flexible, generalizable, and scalable predictive modelling approach for predicting student course performance from ongoing assessment, the inclusion of factors that may impact personal performance in a future learning analytics model could possibly enhance the capability of the machine learning algorithm to extract patterns relating to grades from such data. This can therefore assist institutions in effective course health checks and early intervention strategies, and also modify teaching and learning practices to promote quality education and desirable graduate attributes.

**Author Contributions:** Conceptualization, R.C.D. and A.A.M.A.; methodology, R.C.D. and A.A.M.A.; software, A.A.M.A.; validation, R.C.D. and A.A.M.A.; formal analysis, R.C.D. and A.A.M.A.; investigation, R.C.D. and A.A.M.A.; resources, N.J.D. and R.C.D.; data curation, R.C.D. and A.A.M.A.; writing—original draft preparation,A.A.M.A., R.C.D. and Z.M.Y.; writing—review and editing, R.C.D., S.G., N.J.D., A.D., P.D.B. and Z.M.Y.; visualization, R.C.D., A.A.M.A. and S.G.; supervision, R.C.D.; project administration, R.C.D.; funding acquisition, R.C.D. All authors have read and agreed to the published version of the manuscript.

**Funding:** This research was funded by UniSQ through School of Sciences Quartile 1 Challenge Grant (2018) awarded to R. C. Deo. It was supported by Office for the Advancement of Learning and Teaching under the Technology Demonstrator Project, entitled "*Artificial intelligence as a predictive analytics framework for learning outcomes assessment and student success*". The APC was funded by UniSQ Excellence in Research 2021 grant held by Professor R. C. Deo.

**Institutional Review Board Statement:** The University of Southern Queensland (UniSQ) Human Research Ethics granted ethical approval, enabling the researchers to utilize the examiner return data securely and ethically under approval number H18REA236.

**Informed Consent Statement:** Student consent was waived due to this project being low risk to students, the removal of identifiers, and the strict confidentiality of student records.

**Data Availability Statement:** Examiner records data for student performance in ENM1500 Introductory Engineering Mathematics were made available under by the Faculty of Health, Engineering, and Sciences and the Director of Data Services under Ethics Approval H18REA236.

**Acknowledgments:** The authors thank Associate D. Strunin and Nawin Raj for providing examiner return datasets for ENM1500 Introductory Engineering Mathematics.

**Conflicts of Interest:** The authors declare no conflict of interest.

## Abbreviations

The following abbreviations are used in this manuscript:

| | |
|---|---|
| MARS | multivariate adaptive regression splines |
| KNN | k-nearest neighbour |
| KRR | kernal ridge regression |
| DTR | decision tree regression |
| MOOCs | massive open online courses |
| SVM | support vector machine |
| $GCV$ | generalized cross-validation |
| $BF$ | basis function |
| $MSE$ | mean square error |
| ADNG | Associate Degree of Engineering |
| B.CON | Bachelor of Construction Management |
| $A1$ | Assignment 1 |
| $A2$ | Assignment 2 |
| $Q1$ | Quiz 1 |

| | |
|---|---|
| *Q2* | Quiz 2 |
| *EX* | examination score |
| *WS* | weighted score |
| *RMSE* | root mean square error |
| *MAE* | mean absolute error |
| *WI* | Willmott's index |
| *NSE* | Nash–Sutcliffe coefficient |
| *LM* | Legates and McCabe's index |
| *RRMSE* | relative RMSE |
| *RMAE* | relative MAE |
| $WS_{obs}$ | observed (real) weighted score |
| $WS_{pred}$ | predicted weighted score |
| $U_{95}$ | expanded uncertainty |
| *r* | correlation coefficient |
| $r^2$ | coefficient of determination |
| DR | discrepancy ratio |
| ECDF | empirical cumulative distribution function |
| \|*PE*\| | predicted error |

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
