# Peer review of "Introductory Engineering Mathematics Students’ Weighted Score Predictions Utilising a Novel Multivariate Adaptive Regression Spline Model"

_sustainability, doi:10.3390/su141711070_

Round 1

Reviewer 1 Report

Predicting students' scores in any area may be of great importance for teachers, researchers, and policymakers. This importance augments in fields related to engineering and science in general, as it helps take action on time. Consequently, tools (models) like the one introduced by the authors are relevant to enhance the success rate in any academic course and lower the risk of dropping out due to low academic performance. Therefore, I appreciate the work performed by the authors.

Author Response

Dear Reviewers, 

Please find attached a detailed response to your reviews. 

Sincerely

Professor Ravinesh Deo

Reviewer 2 Report

1. Please add the advantages of your work clearly in the abstract section.

2.  Main them, contribution, and motivation should be highlighted in the introduction section.

3. Significance of your work and how your work is stronger than other existing methods, please justify it.

4. Recent work in the same field in some top-ranking journals must be added in the references section.

5. Improve sentences from 307 to 317 in a good way. Feel some confusion.

6. I think authors should follow a different way in the conclusion compared to the abstract.

7. Please remove all the grammatical errors.

Author Response

Dear Reviewers

Please find attached our response to Reviewer's comments. 

Sincerely

Professor Ravinesh Deo

Reviewer 3 Report

“Introductory engineering mathematics students’ weighted score predictions utilising a novel multivariate adaptive regression spline model”

This paper focus on the prediction of students’ scores based on different sets of information and "aims to develop a student progress-monitoring model that can be used as a vital part of the e-teaching and e-learning systems that guide educators in making better decisions to improve their practice for optimal outcomes". This work is well structured and organized.

In the following are presented some comments and remarks which should deserve the attention of the authors.

Section_Abstract:

This section is well structured and clear.

Section_Introduction:

The contextualization and literature review are clearly defined and explained.

Section_Theoretical overview and Methodology:

Sequentially we observe the MARS as objective model followed by KRR, KNN and ending with DT.

Section Research Context, Project Design and Model Performance Criteria:

The final period of the first paragraph is “equal” to the first period of the second paragraph – lines 223-227.

In lines 236-238 - How were they graded the quizzes, the assignments and Examination? What scales were used?

In lines 263 – 267  the authors refer the distribution of the different assessments. It is needed a deeper explanation of these assessments, ex open questions, multiple choices, …

In lines 283-285 – Detail the idea of this paragraph.

Line 285 – W should be WS ?

In table 1 are presented several descriptive statistics:

-       How do you justify the grade over 1000 and over 100?

-       Why it is not included the median, one of the most important statistics?

In subsection Model Development Stages the authors (line 288) refer that the observation period during 5 years. Why it was considered this global data set and no different and “independent” subsets corresponding to each year?

In table 3 – M3  should be WS=f{Q2}.

In Line 233, the authors mention 21 basis functions. There is a previous explanation on section 2, but here it is necessary a deeper explanation or exemplification of these particular basis functions, because as the authors say it depends on the knots.

Section Results and Discussion

The results and discussion are clearly explained, and the strength of the MARS model is highlighted.

Line 418 – M11.

Section Further Discussion, Limitations of this Work and Future Research Direction

The limitations are well identified which could translate a loss of significance of this study. As it is denoted in lines 534-536 the model should be addressed to each institution.

Section Conclusions

The prediction of the student performance is working in progress every time.

In lines 587-591 – the authors evidence that the MARS model could help understanding the learning needs. In what direction the authors could justify this conclusion?

Author Response

(The authors gave the same response as above.)

Round 2

Reviewer 2 Report

Accepted